# ‘One Health’ Actors in Multifaceted Health Systems: An Operational Case for India

**DOI:** 10.3390/healthcare8040387

**Published:** 2020-10-07

**Authors:** Sandul Yasobant, Walter Bruchhausen, Deepak Saxena, Timo Falkenberg

**Affiliations:** 1Center for Development Research (ZEF), University of Bonn, 53113 Bonn, Germany; Walter.Bruchhausen@ukbonn.de (W.B.); falkenberg@uni-bonn.de (T.F.); 2Global Health, Institute for Hygiene and Public Health (IHPH), University Hospital Bonn, 53127 Bonn, Germany; 3Indian Institute of Public Health Gandhinagar (IIPHG), Gandhinagar 382042, India; ddeepak72@iiphg.org; 4GeoHealth Centre, Institute for Hygiene and Public Health (IHPH), University Hospital Bonn, 53127 Bonn, Germany

**Keywords:** One Health, intersectoral collaboration, actors, health system, India

## Abstract

The surging trend of (re)emerging diseases urges for the early detection, prevention, and control of zoonotic infections through the One Health (OH) approach. The operationalization of the OH approach depends on the contextual setting, the presence of the actors across the domains of OH, and the extent of their involvement. In the absence of national operational guidelines for OH in India, this study aims to identify potential actors with an attempt to understand the current health system network strength (during an outbreak and non-outbreak situations) at the local health system of Ahmedabad, India. This case study adopted a sequential mixed methods design conducted in two phases. First, potential actors who have been involved directly or indirectly in zoonoses prevention and control were identified through in-depth interviews. A network study was conducted as part of the second phase through a structured network questionnaire. Interest and influence matrix, average degree, network density, and degree of centralization were calculated through Atlas.Ti (ATLAS.ti Scientific Software Development GmbH, Berlin, Germany), UCINET (Analytic Technologies, Lexington, KY, USA) software. The identified actors were categorized based on power, administrative level (either at the city or district level), and their level of action: administrative (policy planners, managers), providers (physicians, veterinarians), and community (health workers, community leaders). The matrix indicated that administrative actors from the district level were ‘context setters’ and the actors from the city level were either ‘players’ or ‘subjects’. The network density showed a strength of 0.328 during the last outbreak of H5N1, which decreased to 0.163 during the non-outbreak situation. Overall, there was low collaboration observed in this study, which ranged from communication (during non-outbreaks) to coordination (during outbreaks). The private and non-governmental actors were not integrated into collaborative activities. This study concludes that not only collaboration is needed for OH among the sectors pertaining to the human and the animal health system but also better structured (‘inter-level’) collaboration across the governance levels for effective implementation.

## 1. Introduction

The interaction of humans and animals in their shared environment results in dynamic circumstances in which the health of all is inextricably linked to that of the others [1,2]. Over the last two decades, the surging trend of emerging and re-emerging diseases has been creating significant challenges across the globe [3]. South-Asia is a major hotspot for (re)emerging diseases with India being one of the greatest contributors to the burden of zoonoses [4]. The One Health (OH) approach and its strategies are promoting collaborative actions at the human-animal-environment interface [5,6,7,8], providing opportunities for the prevention and management of zoonoses and guiding zoonoses research and policy. Although the OH approach is not well institutionalized and is facing many challenges in South Asian countries—such as lack of institutional capacity, issues with ownership (lack of mutual interest), each sector have their own mandate, responsibility, priority, and constraints [9]—the emerging outbreaks demonstrate the urgent need for strong coordination and collaboration between the human and animal health sectors to combat zoonotic diseases [3,4,10]. In the case of unforeseeable onset and rapid (re)emergence of zoonotic diseases, the public health system should quickly be able to identify the early signs and react promptly to minimize the threats [11]. This type of situation is the time to embrace an OH approach as a framework for public health action against zoonoses, as indicated by the tripartite (WHO, FAO, OIE) zoonotic guide [12].

Despite having a large number of zoonotic outbreaks, India is one of the other South Asian countries that has not yet implemented a national OH policy and/or operational guidelines. On the one hand, it is indicated that large knowledge gaps on emerging diseases remain in India, due to the lack of data on the social, economic, and public health impact of zoonotic pathogens [13,14,15,16]. On the other hand, zoonotic disease research is largely ad hoc, and the majority of research focuses on the development of vaccines, therapeutics, and diagnostic tests rather than exploring sustainable disease control strategies within the local context [5,17]. Prioritization and appropriate utilization of available resources are of critical importance for the effectual control and prevention of these diseases [3,18]. Multiple actors’ perspectives and participatory actions are important approaches for identifying and implementing sustainable solutions that are adapted to local contexts in consideration of culture and needs [8,19]. In summary, understanding the local context, the collaboration pattern at the human–animal health interface, the detection and response strategies are forming an important foundation for the prevention of zoonotic diseases. The World Bank emphasizes a “more general, permanent system for coordinated national and international surveillance and control” that would entail “more regular channels of collaboration than the current communication between agencies that prevails to date, which is based on temporary arrangements formed in response to various contingencies” [20], therefore highlighting the need to explore the sustainability of existing collaborations and developing strategies to establish sustainable collaborations across sectors.

Exploring and understanding collaboration patterns is a complex process, as the health systems for humans or animals are shaped by informal rules and relations [21,22]. Inter-sectoral collaboration (ISC) is a continuum of varying stages of integration between actors: i.e. communication, cooperation, coordination, collaboration, and coadunation [23]. The different degrees of integration range from fully independent functioning to fully integrated systems [24,25]. Thus, to understand the degree of integration, it is essential to explore its actors at the interface of human–animal–environmental health, their relationships, and interactions [26], which comprises not only the human health system but also the animal health system. It was recognized that multiple actors’ perspectives are essential to understand the level of integration and to address issues like ownership, institutional capacity, or the different mandate of each sector. Research on health system performance indicated the urgent need to understand the actors and institutions along with the formal and informal rules governing the health system at the local context [27].

Therefore, in the absence of an OH policy and/or national operational guidelines, mapping of actors, understanding the existing capacities, and networks in the local context are becoming an important task in India. In India, so far, the zoonoses prevention and control remained under the purview of the division of the zoonotic disease programs under the National Centre for Disease Control, Ministry of Health & Family Welfare for the humans [28]. Whereas the zoonoses among domestic and livestock animals are addressed by the Ministry of Fisheries, Animal Husbandry, and Dairying which is newly formed from the department of the same name under the Ministry of Agriculture and Farmers Welfare in 2019, the Wildlife Institute of India focuses on zoonoses in wildlife [29]. This indicates the fragmented approach to the problem of zoonoses control in the country [13]. The principle of action by these authorities remained as ‘need-based collaboration’: however, the ‘need’ has been documented so far during the outbreak situations only. In absence of national OH policy and/or operational guidelines in India, investigating the presence and distribution of actors for zoonotic disease prevention is required to enable the implementation of OH in the local context. Thus, the overall aim of this study is to identify and categorize actors at the human–animal health system interface and attempted to document the issues and challenges pertaining to the ISC in two different situations (one during an outbreak and another during non-outbreak) with a focus on prevention and control of zoonotic diseases in Ahmedabad, India.

## 2. Materials and Methods

### 2.1. Study Setting

#### 2.1.1. General Setting

India has a quasi-federal form of government, called ‘union’ or ‘central’ government, with elected officials at the union, state, and local levels. The cities of the country rely on the municipal or local governance which refers to the third tier of governance in India, at the level of the municipality or urban local body, and have a great degree of fiscal autonomy and functions, which, however, varies from state to state. In contrast, the rural government system relies on Panchayati raj, a three-tier governance structure with elected bodies at the village, block, and district level.

#### 2.1.2. Specific Setting

This study was conducted in Ahmedabad city of the western Indian state of Gujarat. The Ahmedabad city is selected for two prime reasons: first, it has encountered different zoonotic outbreaks over the past few decades—including Crimean-Congo hemorrhagic fever [30], H1N1 [31], and H5N1 [32]; second, the city has become one of the innovation corners for various governance models [33]. Like other cities of India, Ahmedabad city is governed by a corporate body, the Ahmedabad Municipal Corporation (AMC). The public urban health system for humans relies upon Urban Health Centres (UHCs) and Medical Colleges [34] and is commissioned by the health department of the AMC. The human health system is also enriched by private providers throughout the city [35]. The animal health system is managed by the Cattle Nuisance Control Department with few veterinary dispensaries. In addition, there are few trust (non-profit) agencies and for-profit private facilities also contributing to the animal care across the city.

### 2.2. Study Design

This case study adopted a mixed-method design to collect the information in Ahmedabad, India from September 2018 to October 2019. It is part of a larger health system study called RICOHA (Research to explore Intersectoral Collaborations for the One Health Approach), the detailed study protocol is published elsewhere [36]. In this case study, there were two phases of data collection: phase-I, the qualitative data collection through in-depth interviews (method for objective 2 in RICOHA study protocol); and phase-II, the quantitative data collection through a network survey (method for objective 3 in RICOHA study protocol).

### 2.3. Study Concepts

We have conceptualized OH as a policy or institutional innovation, whose institutionalization process analysis requires a systems approach [37]. We have used the systems approach for two different reasons: first, in a rather theoretical perspective, for understanding OH as a system of functional sub-systems, like in the theory of social systems (by Niklas Luhmann), where the interaction of sub-systems creates new challenges; and second, by seeing social structures as empirically quantifiable constituents of a system, for analyzing the degree of involvement in the collaborative work. This use of two different notions of the system was based on two assumptions. The first assumption was that the ISC process starts with the identification of institutions or groups of stakeholders (defined in social systems theory as sub-systems and here named as collective actors), which are essential to perform the ISC activities on the issues at the human–animal interface and have their own logics and interests. The second assumption was that the development and/or the sustainability of ISC activities might require a simpler or more complex system change across the different sectors [38]. In this case, the social network analysis (SNA) provides the analytical framework for the understanding of actors in the health system and guiding the research process [39].

### 2.4. Study Sample and Sampling

The study was limited to the boundaries of the human and the animal health system, consisting of samples from the top two levels of the health system, i.e., actors from the administrative level (working on planning and decision-making) and actors from the provider level (working in clinical settings and providing healthcare services). As there were very few actors present at the administrative level, we have approached all the actors of the human and the animal health system and most of them provided consent to participate in the study. Thus, from for the qualitative data collection, we recruited almost all the administrative actors working at the AMC and purposively selected the lead non-governmental organizations and private bodies. Similarly, we purposively also selected a few actors from the provider level until the saturation of responses. For the quantitative data collection, in addition to the above participants, we sent an open invitation to all the clinicians working in the human and animal health system of the city and those who agreed to participate (40% response rate) and who provided consent, were recruited for the survey.

### 2.5. Study Data Variables and Data Collection

#### 2.5.1. Phase-I (Qualitative Data Collection)

In-depth interviews were conducted with the actors purposively selected from the administrative and provider level. The one-to-one interviews were done at the date and time convenient to participants after obtaining their consent to participate in the study. An interview guide with broad, open-ended questions on the respondents’ collaboration with other actors during different situations (i.e., outbreak vs. non-outbreak). In addition, a ranking scale was used to collect information on the perceived influence and interest in the prevention of zoonoses activities of different actors. Audio recording and verbatim notes were taken during the interview.

#### 2.5.2. Phase-II (Quantitative Data Collection)

An open invitation to participate in the network survey was sent to all actors from the administrative and provider level of the human and animal health system of Ahmedabad. A structured network questionnaire was administered personally by a trained research assistant to those who responded and provided consent. The demographic information, professional practices, interactions, and collaborative activities (especially across the sectors) were collected as part of the network survey. A list of actors was prepared prior to the survey based on the qualitative findings of the interviews of phase I. These were presented to each participant and all participants were asked to describe their frequency of interaction with each actor on the list on a six-point scale (i.e., no contact, yearly, quarterly, monthly, weekly, and daily). Furthermore, they were asked to describe the degree of ISC with the other actors on a six-point scale from minimal to highest integration (i.e., not linked, communication, cooperation, coordination, collaboration, fully linked). A network tie between actors was defined as at least monthly interaction as frequency and a communication relationship as integration. These relationships were explored under two distinct situations, i.e., during the last outbreak (i.e., H5N1 in 2017) and during the current non-outbreak (at the time of data collection in 2019) situation.

### 2.6. Study Analysis

#### 2.6.1. Phase-I (Qualitative Analysis)

Transcripts from the interview recordings were made on the same day. Content analysis in Atlas.Ti version 8 [40] was used to identify each actor mentioned at least once in the transcript. The need for collaboration was assessed based on themes. The Interest and Influence Matrix (IIM) [41,42] was conducted to understand the presence of actors and their roles in the prevention and control of zoonotic diseases. The IIM categorizes four major types of actors, i.e., players (high interest, high influence), subjects (high interest, low influence), context setters (low interest, high influence), and the crowd (low interest, low influence) [41,42]. Actors with high levels of interest and influence are termed as ‘players’, these are important key elements in the collaboration process. They also help highlight coalitions to be encouraged or discouraged, what kind of decision to be fostered, and also provide information on how to convince other actors. ‘Subjects’ have high levels of interest but low levels of influence. Therefore, although by definition they are supportive, they are unlikely to be able to play a significant role in supporting the implementation. However, by engaging subjects in the implementation process, they might become influential in a later stage by forming an alliance with other influential actors. The actors with high influence but low interest is known as ‘context setters’; however, they might have a significant influence in implementation, but might be difficult to engage in each process. Even sometimes, additional effort is required to engage these actors. It is important to consult these actors for their opinions, concerns, and ideas for successful implementation. The ‘crowd’ are the actors with low interest and influence; thus, little need to consider them in much detail. However, their interest or influence might change over time. It is equally important to inform them of each implementation process.

#### 2.6.2. Phase-II (Quantitative Analysis)

Network analysis was carried out to understand the strength of the interaction between these actors in the outbreak and non-outbreak situations. A visualization of the interactions and quantified outcomes such as average degree (the average number of links each node in the network has), density (the proportion of possible links in the network), and degree of centralization (the extent to which only a few nodes have a large number of ties) were analyzed in UCINET version 6 [43]. The average degree is denoted by the total number of edges or links divided by the total number of nodes in a network. Thus, the value of average degrees depends on the number of actors and their frequency of connections. Similarly, the density is defined as the number of connections a participant has divided by the total possible connections of a participant could have (e.g., if there are 20 people, each person could potentially connect to 19 others, thus if a person were connected to all other persons the density would be 100% (19/19)). The degree of centralization is an indicator of centrality and a good measure of the total number of connections a certain node has, but will not necessarily indicate the importance of each node in connecting to others or how central it is in the network. The values of degree of centralization range from 0 to 1, with 0 indicating no connection and 1 indicating all are highly connected. For the quantitative data, descriptive statistics were created in R version 3.4.1 [44].

### 2.7. Ethics Approval and Consent to Participate

Ethics approval has been obtained from the Research Ethics Committee of Center for Development Research (ZEF), University of Bonn, Germany (ZEF dated 18/06/2018), and the Institutional Ethics Committee of the Indian Institute of Public Health Gandhinagar (IIPHG), India (TRC-IEC No. 02/2018 dated 25/07/2018). The written consent was collected from each participant, who were recruited in this study.

## 3. Results

A total of 30 interviews were conducted as part of the phase-I (12 from administrative, 12 from the provider, 6 from private/non-governmental organizations), followed by 6 actors from the administrative level and 66 actors from the provider level participated in the phase-II.

### 3.1. One Health Actors of the Complex Health System

The presence of two-layered actors—i.e., actors of the local government body (AMC) and actors from the district administrative body—was documented. Although there were similar actors also present at the top level—i.e., at the state or the national level—it was difficult to get information on them from the current interview data. The two layers of actors on the two administrative levels of district and city resulted in a strong influence from the district or even higher authorities on the action of zoonoses prevention in the city. These actors (city, district/higher) have a direct or indirect role in the decision and/or implementation process of the collaborative activities. The actors were broadly categorized by their level of action in the health system: top, middle, or bottom. At the top level, the policymakers, program managers, and planners were considered as ‘administrative actors’, followed by the actors involved in the clinical service provision such as physicians and veterinarians, considered as ‘provider actors’. At the bottom, community leaders, health workers, and non-governmental organizations were considered as ‘community actors’. The analysis revealed that the administrative level held the highest power of influence for zoonoses prevention activities. Although this study was focused on the city level, there were actors from the district or higher authority, who influenced these city-level actors directly or indirectly. This is especially evident during outbreak situations when actors were involved in collaborative activities initiated by the district authority. The administrative actors were acting across the district and city level, as well as across sectors during outbreak situations. Although the governance structure remains the same in the outbreak and non-outbreak situations, the increased influence of the district/higher administrative actors is noteworthy during the outbreak situation. Table 1 represents most of the potential OH actors who have been involved directly or indirectly in the prevention and control of zoonotic diseases of the Ahmedabad city. 

### 3.2. Interest–Influence Matrix (IIM)

The level of interest and the influence on prevention and control of zoonoses was assessed based on the analysis of the semi-structured interviews. Table 1 also represents the actors as ‘players’, ‘subjects’, ‘context setters’, or ‘crowd’ in the study setting. In this case, the city human health administrators, health centers/hospitals, and animal dispensaries/clinics were found to be the key ‘players’, implying that they were strong actors for zoonoses prevention and control with high interest and high influence. However, the city animal health administrators were found to be with relatively low influence although with high interest for zoonoses and are, thus, considered as ‘subjects’. Although ‘subjects’ have low power, there were minimal collaborative activities documented. In addition to the city animal health administrators, the veterinarians, private human health clinics, and community actors, who were assessed as ‘subjects’, need to be strengthened with certain powers. The low influence of the animal health administrators on zoonotic disease prevention and control compared to the human health administrative actors indicates the issue of power rivalry. The overall context was managed by the ‘context setters’, in this case, the top authorities of human and animal health, who influence the overall collaborative activities. There were many actors categorized as ‘crowd’, i.e. with low interest and influence, implying that they were seen as potential actors rather than actual actors, such as NGOs, the city zoo, community leaders, research institutes, private physicians, environment personnel of city and district level, police department, dairy farms, etc.

### 3.3. Issues and Challenges for Intersectoral Collaboration

#### 3.3.1. Perceived Need for ISC

On enquiring about the need for collaboration, most of the actors stressed that outbreaks or health emergencies were the situations during which they require support from other actors. There was no perceived need for ISC activities unless it is directed by the top authorities. Importantly, it was found that collaborative activities only happen after initiation from higher authorities. It was emphasized that these collaborative actions were initiated from the state or national level during outbreaks and that the subordinated actors followed the top-down directive. Otherwise, there was no need for any collaborative actions across the sectors as stroked by the participants. It is, therefore, regarded as necessary by several actors to sensitize all actors about the importance and benefits of collaborative actions to sustain any level of ISC beyond outbreak situations.

“…Our teamwork is not by need; it’s by demand. During the outbreak, the Collector (prime administrative authority of a district) sensitizes all the actors based on the demand for action. And our collaborative effort was very good during the last outbreak”(Human health actor)

“We need stringent collaboration for the diseases which are not reported currently in the system… and all actors need to understand their respective contribution towards the collaborative work…”(Animal health actor)

“We get information on the outbreak alert from state or center and they tell us what to do and how to proceed.”(Human health actor)

In addition, the need for collaboration was only expressed when deficiencies were found within the respective sector. When services and/or resources were required from other sectors and/or beyond the administrative boundary, only then was the need for collaboration emphasized.

“City administration is different and also the city has limited strength for Animal Husbandry, so we wish to collaborate with district officials…”(Animal health actor)

“We (in Human health) have our own system in place and we do have animal husbandry cell at the corporation level. We at AMC meet them (in Animal Health) regularly; however, if we need help like a laboratory or additive human resources, then only we approach the district animal husbandry department.”(Human health actor)

On the one hand, actors at the provider level indicated that physicians only need to interact with veterinarians during emergencies, otherwise it would be a waste of capacities, as most of the practitioners were overloaded with their daily caseload. On the other hand, some of them stated that there was no system in place to interact with cross-disciplinary professionals, so it was never realized.

#### 3.3.2. Challenges for Collaboration

There were different challenges for collaboration highlighted by the actors, one of the major challenges was who is interested or motivated to lead such action. As observed in the local context, collaborations only happen with top-down directives during outbreaks, the power issue that emerges leads to the question of what needs to be done and who should do it in the non-outbreak situations. In addition, the collaborations were perceived as a burden rather than benefits, which is even more problematic than the power issues. Most of the actors did not want to develop ISC as they have perceived it as additional work.

“Within the human health sector, the administrative system is different for the city (urban) and rural…so difficult to collaborate sometime; we directly communicate with the state government regarding any epidemic, outbreak situation…”(Human health actor)

“Animal Husbandry should be the lead for prevention of zoonotic diseases with some support from the human health sectors and transparency is essential for collaboration”(Animal health actor)

“We are in short of human resource, there is a huge shortage of veterinarians and livestock inspectors, with this situation how to collaborate with other sectors…; I am afraid it would increase the burden on our department”(Animal health actor)

Challenges such as information flow, disease-reporting patterns, knowledge gaps, limited resources, and awareness level were among other challenges for collaborative work. Within the human and animal health system differences in the pattern of information flow and disease, reporting was reported by the participants. Most actors agreed that in the absence of a structured guideline indicating who is to take on which role, collaboration is not possible. As all collaborations were based on specific instructions from the top authorities during an outbreak situation, the actors could not visualize any form of collaboration during non-outbreak periods or see the need for such collaboration. Nonetheless, some made recommendations on how to develop collaboration in the local context if needed.

“All staffs need to undergo training on the need of collaborations for zoonoses disease management, prevention, control through a common platform at the city level including the private actors”(Human health actor)

“Circular training is essential for the front line health workers, who never studied what zoonoses are! If we train and sensitize our multipurpose healthcare workers, then they could also work on zoonoses prevention, as they have a good reach to every house of the community”(Human health actor)

“Whatever and however we collaborate, if people will not (be) aware enough then prevention of any zoonoses will be difficult, sometimes we provide awareness without the help of a medical doctor…and media may play a vital role in sensitization”(Animal health actor)

#### 3.3.3. Continuing Neglect of Private Actors in Collaborations

Collaborating with the actors from the private sectors was not evident in the local setting. Most private and non-governmental actors were neither involved in any collaborations nor contributed to zoonoses prevention significantly. However, some non-government actors, i.e., animal welfare organizations, were working with the AMC on activities such as animal birth control, census, etc.

“Non-governmental organizations are great helping hands in livestock care, so we should strengthen their effort by providing further training and educating them on various preventive activities.”(Animal health actor)

“We (NGOs) do not get any support (neither financial nor technical) from Govt., so why we will collaborate with them?”(NGO actor)

“Govt. never ask us (private providers) to collaborate for anything, I am trained abroad and I can contribute in many things, but Govt. never provided a scope to work with them….”(Animal health actor)

“Private practitioners are never prioritized to be part of the health system, although we contribute largely to the healthcare and also there is no guideline for involving private actors, thus we lack cooperation!”(Human health actor)

### 3.4. Interconnectedness of the Actors in the Health System Network

The health system network analysis provided quantitative support to the qualitative findings and compared the interaction among the actors during the last outbreak (H5N1 in 2017) with those of non-outbreak situations. The analysis of different network parameters for both outbreak and the non-outbreak situation are shown in Table 1. A density value of ‘1’ is expected in a fully collaborated network. The overall network density signified higher interaction among the actors (0.328) during the outbreak as compared to the non-outbreak situation (0.163). This pattern was also observed upon disaggregating the data by the health system level (see Table 2). The other two network parameters—i.e., average degree and degree of centralization—have also exhibited higher values during outbreak compared to non-outbreak situations. The degree of centralization signified that few nodes have higher ties, especially among the administrative actors. Although the degree of centralization among the administrative actors (0.564, 0.473) and provider actors (0.625, 0.607) remained the same in the outbreak and non-outbreak situation respectively, the degree reduced (from 0.556 to 0.205) among the interaction between administrative and provider actors. This highlights that cross-level interaction varies greatly between outbreak and non-outbreak situations. At the same time the higher degree of centralization, which remained the same across outbreak and non-outbreak situations, indicated that few actors govern the collaboration pattern and information flow and are; therefore, the key actors to establish sustainable collaboration patterns. Figure 1 presents the nodes and their ties in the two discussed situations. For visualization purposes, different shapes and colors were used: dark-colored squares for administrative actors, medium grey diamond shapes for provider actors, and light grey circles for community actors. One important finding was that the prime administrative actors, who were well-positioned and highly interconnected during outbreaks, significantly reduced the number of ties during non-outbreak situations. The IIM matrix also reflected that the district level administrative actors have a high influence on most city-level actors, which resulted in coordinated activities during the outbreak. These density values could not be attributed directly to the stage in the continuum of ISC, nonetheless, a qualitative attribution indicates a range from the communication (during non-outbreak) to the coordination (during an outbreak) as per the need in the local context. In summary, there was significantly lower interactions during non-outbreak situations across all the network measures, very low ISC both during the outbreak and non-outbreak situations; however, a high centrality remained at the administrative level both in the outbreak and non-outbreak situation.

Figure 1a,b represents the network cohesion during the outbreak and non-outbreak situation. The lines between actors represent their respective interactions. Whereas an arrow pointing away from an actor towards another actor shows the former mentioned the latter, a line with arrows at both ends is a mutual relationship. Table 3 represents the collaboration details of 66 medical officers, physicians, and veterinarians who participated in the network survey. As there were few actors from the administrative level, no such descriptive analysis was conducted. Among the 66 actors from the provider level, there were 74% and 26% belonging to human health and animal health respectively. Most (84%) were held bachelor degrees and had 12 ± 8 mean years of professional experience. One quarter (26%) were working in the private or non-governmental sectors. Among the participants, only 27% had collaborated with other actors during the outbreak situation. Among the rest, reasons given for non-collaboration were that collaboration among physician and veterinarians were not at all required (58%), followed by lack of any policy/guidelines (24%), and lack of knowledge (18%).

Upon exploring their viewpoint on the potential actors, who could bridge between the human and animal health system, the majority of the participants at the provider level (64%) mentioned that actors from the community level could act as such bridging actors by identifying symptoms at the early stage and reporting to the corresponding authorities. This signified that the preventive actions need to be shifted towards the bottom of the health system rather than developing collaborations at the administrative and/or provider level. Formal training on any zoonotic diseases was found to be absent in 40% of participants although about 65% participated directly in the health campaigns related to zoonoses. In summary, a low perceived need for collaboration was voiced at the provider level further indicating that collaboration is perceived as a burden, thus these provider actors urged to shift the support of ISC to the bottom level.

## 4. Discussion

This study identified actors from various levels of the health system in the local setting of Ahmedabad for the prevention and control of zoonoses. Although the city level actors were having prime actions, there was another layer of administrative actors from the district or the top authorities found to influence the city actors. During outbreaks, the top authorities from the district and/or state directed the collaborative work with the city actors. This type of collaborative activities has been documented in the literature [45,46,47], where actors have to perform or enable interdisciplinary work in addition to their specific roles and responsibilities of the routine work. Thus, a firmer borderline between the actors of the city and higher authorities (governance structure) could not be drawn in the collaborative activities, as ISC also does not intend to do so. It is indicated that local government bodies are often weak and powers devolved to them with limited resources might have resulted in being neglected or overlooked on issues of zoonoses [48,49]. In the current study, the district authorities were superimposed over the city authorities for the collaborative actions during health emergencies, while the city actors were also found to be dependent on the district authorities for the required resources, especially skilled human resources. This type of resource dependence between these two layers of actors (i.e., state vs. city level) was also observed in other healthcare settings of the country [50,51]. Although the OH approach has especially emphasized the need to promote ISC among the human and animal sectors [12,52], we found that collaboration across administrative levels, across state and city sectors, are also of utmost importance. One of the international committees—i.e., Federation of Veterinarians of Europe (FVE)—and the Standing Committee of European Doctors (CPME) also emphasized that collaboration across all levels is essential for the operationalization of OH [53]. In this case, the city actors should not be isolated from other actors (district/top-level) during the operationalization process, as high inter-dependency was observed. Furthermore, this study identified that private and non-governmental actors had minimal or no interaction with the governmental actors. Yet, there are several public-private partnership programs evinced in strengthening the health system globally [54] or nationally [55] and the importance of private actor engagement in reducing the disease burden in India is documented [56,57]. One Health Network analysis by Spencer J. et al., indicated the minimal engagement of private for-profit sectors as part of the OH [58]. In global health governance, the private and non-governmental actors contributing significantly towards improving the health for all [59]. Therefore, private and non-governmental actors should not be neglected in developing ISC for OH implementation.

This study documented very low interest in ISC across all actors and that any form of collaboration depended on top-down directives. The operationalization of OH is, therefore, difficult to realize in Ahmedabad, unless all actors are sensitized about the advantages of collaborative actions and clear top-down regulations are developed. Challenges, such as differences in working culture, lack of skilled human resources, and capacities of the respective sectors leads to the shifting of responsibility for zoonoses to other authorities and make ISC a perceived burden rather than an advantage in Ahmedabad. As some of these operational challenges have been documented in the literature [60,61,62,63], it is recommended to sensitize the actors first about the advantages of ISC in the local context by utilizing the current health system network. Degeling et al., also pointed out political and legal issues in the OH decision-making processes and that these should be considered in the local context during operationalization [64]. While it was evinced that administrative actors do work collaboratively during health emergencies or disasters [65,66], they also tend to collaborate with required actors from different levels for their routine work [67]. However, in this study, collaborative work has been documented during outbreaks only, with minimal interactions being sustained during non-outbreak periods among the administrative actors. Although this collaboration was only possible because of the top-down directive, either from the state or the higher authorities, the actual collaborative actions took place at the grass-root level with the help of these administrative actors. This signifies the leadership capacity of the city administrators and could provide opportunities for strengthening and operationalizing ISC during non-outbreak situations [68]. It is indicated that leadership and managerial skills are essential for OH capacity development [69,70], which is recommended to strengthen ISC at the administrative level.

The collaboration at the provider level, especially between physicians and veterinarians, were found to be very poor. However, there is some supportive global evidence that OH collaboration at the provider level is possible through case referral systems or combined OH clinics [71,72]. A study by Speare et al., explored the possibilities for physicians and veterinarians to formally collaborate in managing zoonoses in clinical situations and documented that 90% of actors agreed to collaborate if the appropriate insurance covers the cost [73]. As this is not the case in this study, some sensitization through OH training would be recommended. Another review also discusses the role of veterinarians and physicians in confronting zoonoses and discusses the importance of multi-sector partnerships in controlling zoonoses [74]. Furthermore, this type of ISC has also been envisaged in developed nations, where physicians and veterinarians are involved in joint patient counseling or joint clinical services for minimizing the risk of zoonoses [75,76,77]. In the local context, the actors from the provider level indicated that ISC should be at the community level rather than at the provider level. This might be due to the high caseload in their daily routine, absence of clinical knowledge on zoonoses, or lack of awareness. In the local context, the horizontal collaboration between the human and animal health system at the community level is not possible, as there are no active actors from the animal health system; however, the existing network of community health workers of the human health system could be utilized for raising the community awareness and risk mitigation. Similar community-based risk mitigation through an OH approach has been documented in other parts of the globe [78]. The network cohesion among the provider level actors did not differ much between the outbreak situation and the non-outbreak situation. One of the potential reasons could be the presence of private veterinarians in the network, and the other reason could be few of the provider actors were only engaged in collaborative activities during the outbreak resulted in the minimal difference in the network cohesion at the provider level.

In absence of studies on how to explore the ISC for OH, this study is the first of its kind to recommend the methodologies that have been used in this exploration process. A mixed-method of exploration is beneficial in terms of assessing the situation and understanding how and why in a glance. The qualitative interviews in this study provided an insight on the potential actors of OH as well as the issues on ISC and the same time the quantitative network survey provided the measurements on the network cohesion in the outbreak and non-outbreak situation. Furthermore, exploring the health system at two different levels—i.e., at the administrative level and the provider level—indulged a comprehensive scenario for the OH. Therefore, this mixed-method could be used in any other setting not only within India but also in any other setting across the globe for exploring the OH actors and their network cohesion to understand the ISC.

Given the low interest and lacking perception of a need for ISC in Ahmedabad, coupled with the consensus that top-down directives are required for any collaborative activities, developing such top-down guidance for the administrative actors along with extensive training on ISC at the provider level is urgently needed for operationalizing OH in the context of Ahmedabad [20,79]. Globally, it has been highlighted that a key challenge for promoting sustainable ISCs is lacking political commitment, competition between bureaucratic agencies, different governance structures across sectors, and lacking a common understanding of collaboration across actors [80,81]. Similar challenges for establishing sustainable collaborations were identified in this study for the context of Ahmedabad: lack of interest, lack of resources, lack of political commitment, and lack of guidelines. To establish successful ISC, effective communication and advocacy among the actors for both levels are required. Furthermore, to tackle the issue of low interest for ISC, structured joint training programs (especially for the provider level actors), documentation of success case stories around the country, and evidence on cost–benefit-ratios could help in promoting ISC in the local context. Additionally, guidance documents need to be developed jointly among the actors to establish a framework for ISC in the context of zoonotic disease prevention and control.

### Limitations

It is important to note that the study was undertaken in one city of India and that the generalizability of the findings to any other part of the country might be constrained as cities of India have different governance structures. However, the approach used in this study could be utilized in any context for exploring OH actors. There are certain limitations of the study. First, this study conducted the health system network survey with a cross-sectional design; however, a longitudinal design might have provided better insights into the dynamics of the relationships during different situations. Especially the considerable time since the outbreak in 2017 might have induced recall bias. Second, this particular study did not capture the viewpoint of the community leaders and other potential actors at the community level, the interactions were captured based on the responses of the administrative and provider actors. Thus, it is recommended to consider the perception of leaders and other actors in the community in such a network analysis for improving operational policies. Third, this study did not include the governance process of interaction among the actors within the respective system, which might be of importance as the policy or guidelines and working culture is distinctly different in the human and the animal health system.

## 5. Conclusions

This study identified the presence and analyzed the interaction of the OH actors in the context of Ahmedabad, India, who are directly or indirectly involved in zoonotic disease prevention and management. The actors at the district level who act as ‘context setters’ over the ‘players’ and ‘subjects’ at the city level were found to be the drivers for collaboration. The collaboration strength, in the form of network density, decreases between outbreak and non-outbreak situations. With minimal ISC at the provider level and in the absence of community workers of the animal health system, the physicians and veterinarians recommended strengthening the ISC at the community level by vertical programs and top-down directives. Some of the major challenges that have been identified in this study were the lack of interest for ISC, low perception of any advantages of ISC, and lack of political commitment to ISC. The hitherto minimal involvement of the private and non-governmental actors needs to be enhanced. As most of the actors are relying on the top-down directives, the necessary policy/guidelines on zoonoses prevention with a focus on the OH approach and ISC are recommended to be developed. Keeping the differential governance structure and power rivalry in mind, one idea could be a One Health Task Force of Ahmedabad (OHTFA) with representative actors from the administrative, provider, and community level across the OH domains (human health, animal health, environment) and also including private actors. The design, duties, and powers, working modus, and acceptance such a OHTFA would have to be the objectives of a further study. The last conclusion concerns the lowest level of public action: the presence of community actors from the human health system should be considered as an advantage for awareness-raising on zoonosis in the local context. 

## Figures and Tables

**Figure 1 healthcare-08-00387-f001:**
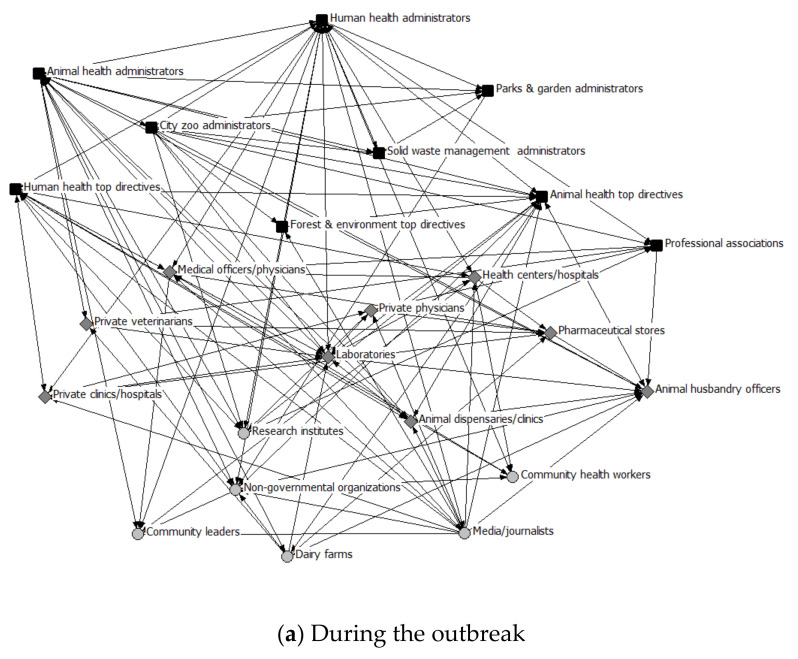
Network view of actors (**a**) during the outbreak and (**b**) during the non-outbreak situations in Ahmedabad, India. Administrative actors: dark-colored squares; Provider actors: medium grey diamond shapes; and Community actors: light grey circles.

**Table 1 healthcare-08-00387-t001:** OH actors for zoonotic disease prevention and control in Ahmedabad, India segregated by level of action and with status in the interest–influence matrix.

Health System Level	One Health Actors	Status in the Interest–Influence Matrix (IIM)
**Administrative level ^$^**	Human health administrators ^#^ Animal health administrators ^§^ Parks & Gardens administrators Solid waste management administrators Professional associations City zoo administrators	Player Subject Crowd Crowd Context setter Crowd
**Provider level**	Health centers/hospitals Medical officers/physicians Private clinics/hospitals Private physicians & infectious disease specialists Nurses/Mid-Wives Pharmaceutical stores Laboratories Animal dispensaries/clinics Government veterinarians Private veterinarians Livestock inspectors/Animal workers	Player Subject Subject Crowd Crowd Crowd Crowd Player Subject Subject Crowd
**Community level**	Community health workers Non-governmental organizations Community leaders Research institutes Media/journalists Households and community Dairy farms Police	Subject Crowd Crowd Crowd Crowd Crowd Crowd Crowd

^$^ Influenced by the actors from the top directives such as district/state human health administrators (Context setter), animal health administrators (Context setter), forest and environment administrators (Crowd). ^#^ Consists of chief medical officer of health, deputy health officer—epidemic, nodal officer of National Urban Health Mission, assistant health officer—entomologist, deputy health officers (zonal level). ^§^ Consists of the superintendent of the Cattle Nuisance Control Department, inspectors.

**Table 2 healthcare-08-00387-t002:** Network strength of actors disaggregated by the health system level in the operational setting of Ahmedabad, India.

Network Measures	Sub-Groups	During Outbreak			During Non-Outbreak		
		Admin	Provider	Community	Admin	Provider	Community
Average degree	Overall	2.652			1.406		
	Admin	5.833	3.001	1.500	2.667	1.417	0.417
	Provider		3.2222	0.556		3.111	0.222
	Community			1.800			0.600
Network density	Overall	0.328			0.163		
	Admin	0.530	0.333	0.136	0.242	0.157	0.038
	Provider		0.403	0.069		0.389	0.028
	Community			0.450			0.150
Degree of centralization	Overall	0.424			0.257		
	Admin	0.564	0.556	0.382	0.473	0.205	0.173
	Provider		0.625	0.232		0.607	0.125
	Community			0.183			0.167

**Table 3 healthcare-08-00387-t003:** Characteristics and collaboration details among the actors at the provider level especially physicians and veterinarians of Ahmedabad, India.

Variables						*N* = 66 (%)
Type of provider						
				Human health		49 (74.2)
				Animal health		17 (25.8)
Gender						
				Male		43 (65.2)
				Female		23 (34.8)
Education						
		Bachelor degree (MBBS/BVMS)				55 (83.7)
		Specialist (MD/MVM)				11 (16.6)
Total years of professional experience (years)						12 ± 8
Work setting						
	Government					49 (74.2)
	Private/Non-governmental					17 (25.8)
Ever involved in inter-sectoral collaborative activities						
			Outbreak management			18 (27.3)
			Advocacy/Administrative			11 (16.7)
Reasons for lack of collaboration						
				No policy/guidelines/opportunity		16 (24.2)
				Lack of knowledge		12 (18.2)
				Not at all required		38 (57.6)
Potential actor who can bridge the human and animal health system						
					At the administrative level	19 (28.8)
					At the provider level	28 (42.4)
					At the community level	42 (63.6)
Ever received any training on zoonoses						39 (59.1)
Ever attended health campaigns related to zoonoses						43 (65.2)

## Data Availability

Data from this study will be available at the Center for Development Research (ZEF), Bonn, Germany, after the completion of this study. Researchers who meet the criteria for access to confidential data are encouraged to approach Timo Falkenberg, Coordinator Fortschrittskolleg ‘One Health’, Center for Development Research (ZEF), Bonn, Genscherallee 3, 53113 Bonn, Germany. Email: falkenberg@uni-bonn.de.

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
