# Peer review of "‘One Health’ Actors in Multifaceted Health Systems: An Operational Case for India"

_healthcare, 2020, doi:10.3390/healthcare8040387_

Round 1
Reviewer 1 Report
Overall, this is a well-written article trying to contribute to an important topic. The findings fit the knowledge we have regarding collaboration activities to implement One Health strategy. I would suggest the following minor revisions.
- Firstly, I found it is quite hard to follow with the different actors in figures 1, 2, and 3. My suggestion is to make a table of actors and their abbreviations before showing any of the figures. Also, keep the number of actors consistent on each of the figures.
- I do not see the necessity of having 6 figure legends for Fig1 and 2. I recommend to only keep 2, that is, Human Health and Animal Health.
- Table1, more explanation needed for “Average degree”. Assuming “Degree of centralization” is at full scale if it is 1, same as the definition provided in line282 for “Network density”, what is the definition and scale of “Average degree”?
- Table1 and paragraph 3.4., I think there should a measurement, i.e. p-value when talking about significant differences.
- A question related to Table 2, it would be interesting to provide the collaboration details of at the community level. I assume enough interviews were done at this level.
- A related question to the citation #25 (2018 paper), which has the details of study design for the current paper (line 81). Please make it clear which part of the 5 method sessions (2018 paper) this paper refers to. For example, is the “method for object2” in the 2018 paper equivalent to Phase I in the current paper?
Author Response
Thanks for your valuable time in reviewing this manuscript. We are thankful for your comments and suggestions to improve this draft. We tried our best to address the specific comments as mentioned in the table below-
|
Reviewer comments |
Author’s response |
Page & Line No. |
|
Firstly, I found it is quite hard to follow with the different actors in figures 1, 2, and 3. My suggestion is to make a table of actors and their abbreviations before showing any of the figures. Also, keep the number of actors consistent on each of the figures. |
Thanks. We have removed the figure 1 & 2 and transferred all the information to the table 1. We hope the table.1 is now intuitive and informative. |
P6. Table 1 |
|
I do not see the necessity of having 6 figure legends for Fig1 and 2. I recommend to only keep 2, that is, Human Health and Animal Health. |
We have removed the figure 2 and the information was transferred to table 1 for ease of understanding |
P6. Table 1 |
|
Table1, more explanation needed for “Average degree”. Assuming “Degree of centralization” is at full scale if it is 1, same as the definition provided in line282 for “Network density”, what is the definition and scale of “Average degree”? |
Thanks for the suggestion. We have now described the parameters of the network analysis in the method section. |
P5. L36-45 |
|
Table1 and paragraph 3.4., I think there should a measurement, i.e. p-value when talking about significant differences. |
UCINET software could not generate the CI for these parameters. The term significant should not be used here. Thus, we have deleted the term significantly. |
P10. L13 |
|
A question related to Table 2, it would be interesting to provide the collaboration details of at the community level. I assume enough interviews were done at this level. |
Thanks for the suggestion. We have collected information only at the administrative and provider level. |
No change |
|
A related question to the citation #25 (2018 paper), which has the details of study design for the current paper (line 81). Please make it clear which part of the 5 method sessions (2018 paper) this paper refers to. For example, is the “method for object2” in the 2018 paper equivalent to Phase I in the current paper? |
Thanks for the great suggestion. Now we have added this information- In this case study, there were two phases of data collection: phase-I, the qualitative data collection through in-depth interviews (method for objective 2 in RICOHA study protocol); and phase-II, the quantitative data collection through a network survey (method for objective 3 in RICOHA study protocol). |
P3. L42, P4. L1-2 |

Reviewer 2 Report
Comments:
Collaboration, particularly between human health and animal health sectors, is a key element of the One Health approach. This study introduces a new method to examine the extent of this collaboration, and to identify some of the barriers to collaboration, through a case study in a city of India. While the findings are of relevance to the context, of more interest is the method, and what advantages / or disadvantages this method has. One of the key issues is the lack of sufficient description of the institutional and governance context of the study – yet it would appear that issues of vertical authority are some of the key factors in impact on ISC.
Given that the study is as much a test of the methodology as of the specific context, more emphasis on the strengths and limitations of the method could be introduced into the methods and discussion. How has this approach differed from other studies of one health collaboration ? what did the researchers learn about the process of implementation of this method ? How do the findings compare with other studies of one health collaboration ?
There are some elements of the method that are not described – such as the number of informants at each level; the response rate, and whether and how many respondents declined to be interviewed; the representativeness of the respondents from across the sectors and professional / administrative groups that participated. These issues need to be addressed. More description is also required of the context, noting that the context is an important factor – in particular the institutional governance arrangements, particularly as these seem to be important factors in influencing the extent of collaboration.
A particularly interesting finding is that collaboration was greater during an outbreak, than in a non-outbreak period. This is an important finding but it is not clear as to what factors in the context between outbreak and non-outbreak periods may be responsible for these differences.
Abstract
Rationale could be clearer – which actors
Little on context
What are implications of findings for OH ? This could include some of the key comments in the conclusions.
Introduction
L 42 Expand on the challenges faced in S. Asia; why is coordination and collaboration an issue ? Provide a definition / explanation for OH and how this relates to zoonoses.
L 46 Rationale for introduction of India ?
L46-59 Rather theoretical – Line 54 context and collaboration as prevention of zoonotic diseases - but also required for the detection and response
‘Sustainable collaborative system’ is vague: be more specific about what collaboration is required, at what levels and how this relates to OH and zoonoses.
L 60 ‘the system’ – what system ?
L 65 ‘health system’- but OH extends beyond the ‘human’ health system to broader collaboration – can you define the extent of the ‘system’, and what actors you expect to collaborate ?
L 67 ‘issues of leadership’ – need to explain where / how leadership relates to integration / collaboration
L 71 suddenly introduces ‘in India’ – some explanation of why the interest in India; what are the specific issues in India. What is meant by ‘OH not yet been institutionalized in India’
As context is important, more information is needed on the context of OH in India – in terms of the regulatory / policy framework; institutions involved; activities in which collaboration is expected to occur
L 73 statement of aim ; what is meant by the ‘health system network’ and how does this relate to collaboration ? who are the actors involved ?
Materials and methods
L 78 Location
It would be helpful to move 2.3 study setting to commence this section – with the additional of the rationale for the selection of this city. to what extent is this city typical of the Indian context ?
It would be helpful to provide more information on the governance organization in India – the federal system; the relationships between cities and states; the authorities and responsibilities of each level of government; and to describe the institutions and sectors that are the subject of the study
L 86 – ‘systems approach’ . There are many ways in which a ‘systems approach’ could be applied – please provide more explanation of what you mean here by a ‘systems approach’ – is this an analytic approach; or does it relate to understanding OH as a system ?
L 104 – ‘samples’ – I presume you mean here the sampling for the interviewees ? The process of sampling is not described – what was the basis for the ‘purposive’ sampling; how large is the study population; how many were sampled; and how were those sampled identified ?
L 120 – how was the structured network questionnaire administered ? by interview or was it transmitted in some way ?
There is no mention of ethical review or ethical procedures in the data collection – although tis is noted in the notes at the end of the paper.
Results
Please provide a summary of the response rate and the key features of the informants – what positions; what sector
L 150 – what is meant by ‘two-layered’ actors ? do you mean that actors were based in one of two layers – the AMC and the district administrative body ? what is the relationship between these bodies ? A diagram of the institutions involved and their relationships would assist in understanding. This would assist in understanding this paragraph. Figure 1 does not clarify the organizational relationships.
See comment above on providing more on the governance / organizational context.
In what ways does the ‘outbreak’ situation differ from the non-outbreak situation – particularly in terms of governance and distribution of authorities ? Does an outbreak declaration change the distribution of authority ? This might help understand the differences found here.
3.2 Interest-influence matrix
The matrix could be described in the materials and methods section, as well as what is meant by the differentiation between players, subjects, context setters or crowd. This would assist in understanding this section.
Line 185-186 – This comment and other comments what is ‘needed’ eg ‘ need to be strengthened with certain powers’ are more of an opinion / interpretation rather than a finding, and belong in the discussion section
Line 191-192 – Again a comment this is more an interpretation / recommendation – and should be shifted to the discussion.
Line 232 – ‘ power issues’ – please explain what is meant by this term. The next sentence talks about ‘who would lead’ – is this a question of ‘who has the authority to lead ?’ or is it a question of who is interested / motivated to lead ?
Given the differences between outbreak and non-outbreak situation mentioned above – do these power issues relate to both situations ? or more in one than another ?
Line 234 – what sort of ‘collaborative activities’ – what sort of collaborative activities are meant here ?
Line 247 ‘were observed’ – by whom ? by the researchers or by the informants ? The comment regarding ‘need to streamline’ – is that a finding or an opinion of the researchers ? If so should be moved to the discussion.
3.4 Network analysis
The parameters discussed here and the analytic approach should be presented in the methods / materials and explained eg the degree of centralization and the average degree; the network figures – at least in broad terms.
Line 289 Are there confidence intervals defined for these parameters ? how significant are the differences between the parameters between outbreak and non – outbreak situations ?
Line 290 ‘reduced significantly’ – is this statistical significance or some other measure ?
Line 295 / Figure 3: please provide some explanation for the figures – what do the line s and the position of the different actors indicate ?
Line 315: This is the first mention of the number of informants – as mentioned above, the response rate should be included at the start of the results section.
Line 334 : was the ‘urging to shift ISC to the bottom level’ an opinion expressed by the informants, or is this the opinion of the researchers ?
Discussion
Line 340: refers to prevention and control of zoonoses, rather than OH. Need to be consistent on whether this is about OH or zoonosis control.
Line 342: ‘integral role in decision making’ – there is no mention of decision making in the results – need to justify this statement.
Line 346: what is the ‘firmer borderline’ and how does this relate to ISC ?
Line 349 – ‘powers devolved to them’ – there is no reporting of the powers devolved to local government bodies in the results section
Line 352 – ‘dependent for required resources’ – again there is no report of this in the results section
Line 358-359 – the issue of vertical collaboration (across levels) is important, but is only raised at this point. It could be mentioned in the introduction and methods as one of the elements for the analysis.
Line 366 – while HSS may be ‘everybody’s business’, this research was not about HSS but about OH collaboration – so the relevance of the private sector to OH should be the basis for their involvement.
Line 380 – the distinction between outbreak and non-outbreak contexts is a key finding – more analysis of what changes in the context and in the relationship[s between outbreak and non outbreak situations would be a useful addition
Line 384 ‘their leadership capacity’ – whose leadership capacity ?
Line 389 ff: useful discussion on provider level – are there differences at this level between outbreak and non outbreak situations ?
Line 407 ff : these are a series of recommendations in how to improve ISC – and could include here some of the references to what is needed provided elsewhere in the paper. It would strengthen the recommendations to provide references as to where the proposed actions have been found to be effective esp lines 416 ff.
Line 422 Limitations. It might be useful to note that the study was undertaken in one city, and that the generalizability of the findings needs to consider that specific context. Given these limitations, here generalizable are the findings ?
There is no discussion of how the findings of this study compare with other studies of OH collaboration either in India or elsewhere.
Line 444 the recommendation for a OH taskforce is interesting – but only arises in the conclusion section. The study did not explore this option – at least it was not reported – so the basis for this recommendation is not clear.
Author Response
Thanks for your valuable time in reviewing this manuscript. We are thankful for your comments and suggestions to improve this draft. We tried our best to address the specific comments as mentioned in below table-
|
Content |
Reviewer comments |
Author’s response |
Page & Line No. |
|
Abstract |
Rationale could be clearer – which actors |
Actos across the domains of OH |
P1. L14-15 |
|
Little on context. What are implications of findings for OH ? This could include some of the key comments in the conclusions. |
We have added a conclusive statement to the abstract now- This study concludes that not only ISC is needed for OH among the sectors pertaining to the human and the animal health system but also better structured (‘inter-level’) collaborations across the governance levels for effective implementation. Within the conclusions we already emphasized and now further clarified the need for new vertical programmes and top-down directives with new tasks for activists at the community level and support as well as inclusiveness at the provider level. |
P1. L31-34 P1. L18-19. |
|
|
Introduction
|
L 42 Expand on the challenges faced in S. Asia; why is coordination and collaboration an issue ? Provide a definition / explanation for OH and how this relates to zoonoses. |
Thanks for pointing this out. We have added now certain challenges that have been identified in the SEA setting such as lack of institutional capacity, issues with ownership (lack of mutual interest), each sector have their own mandate, responsibility, priority and constraints Also we have now extended the section on explaining OH & zoonoses (In the case of unforeseeable onset and rapid (re)emergence of zoonotic diseases, the public health system should quickly able to identify the early signs and react promptly to minimize the threats. This type of situation is the time to embrace OH approach as a framework for public health action against zoonoses, as indicated by the tripartite (WHO, FAO, OIE) zoonotic guide. |
P2. L5-6, L7-12 |
|
L 46 Rationale for introduction of India ? |
We have added a sentence now for India (Despite having large number of zoonotic outbreaks, India is one among other South Asian countries has not yet implemented the national OH policy and/or operational guidelines. ) |
P2. L13-14 |
|
|
L46-59 Rather theoretical – Line 54 context and collaboration as prevention of zoonotic diseases - but also required for the detection and response |
Yes, now we have added the same in Line 23 |
P2. L24 |
|
|
‘Sustainable collaborative system’ is vague: be more specific about what collaboration is required, at what levels and how this relates to OH and zoonoses.
|
Thanks. We have added on what World Bank emphasizes ‘more general, permanent system for coordinated national and international surveillance and control’ that would entail ‘more regular channels of collaboration than the current communication between agencies that prevails to date, which is based on temporary arrangements formed in response to various contingencies’ |
P2. L24-28 |
|
|
L 60 ‘the system’ – what system ? |
Health systems of human or animals |
P2. L31-32 |
|
|
L 65 ‘health system’- but OH extends beyond the ‘human’ health system to broader collaboration – can you define the extent of the ‘system’, and what actors you expect to collaborate ? |
We have explained now- at the interface of human-animal-environment, their relationships and interactions, which comprises not only the human health system but also the animal health system. |
P2. L36-38 |
|
|
L 67 ‘issues of leadership’ – need to explain where / how leadership relates to integration / collaboration |
We have removed the term leadership and replaced with the issues like ownership, institutional capacity, different mandate of each sector. |
P2. L38-40 |
|
|
L 71 suddenly introduces ‘in India’ – some explanation of why the interest in India; what are the specific issues in India. What is meant by ‘OH not yet been institutionalized in India’ |
We have replaced the word institutionalized with the in absence of OH policy and/or national operational guidelines |
P2. L43-45 |
|
|
As context is important, more information is needed on the context of OH in India – in terms of the regulatory / policy framework; institutions involved; activities in which collaboration is expected to occur |
Thanks. Now we have added the actors and policy information to this section |
P2. L43-45 to P3. L1-9 |
|
|
L 73 statement of aim ; what is meant by the ‘health system network’ and how does this relate to collaboration ? who are the actors involved ? |
We have now edited the aim for more clarity. The the overall aim of this study is to identify and categorize actors at the human-animal health system interface and attempted to document the issues and challenges pertaining to the ISC in two different situations (one during the outbreak and another during non-outbreak) with a focus on prevention and control of zoonotic diseases in Ahmedabad, India. |
P3. L12-13 |
|
|
Materials and methods
|
L 78 Location It would be helpful to move 2.3 study setting to commence this section – with the additional of the rationale for the selection of this city. to what extent is this city typical of the Indian context ? It would be helpful to provide more information on the governance organization in India – the federal system; the relationships between cities and states; the authorities and responsibilities of each level of government; and to describe the institutions and sectors that are the subject of the study |
Thanks, As suggested we have moved this section up as mentioned as 2.1. Study setting. We have now also added further information on the governance structure of the city including its functionality mentioned in the section 2.1.1. General setting (general structure in India) and section 2.1.2. Specific setting (about the study site)
|
P3. L16-36 |
|
L 86 – ‘systems approach’ . There are many ways in which a ‘systems approach’ could be applied – please provide more explanation of what you mean here by a ‘systems approach’ – is this an analytic approach; or does it relate to understanding OH as a system ? |
Both. We have used the systems approach for two different reasons, first in the way of the theory of social systems (by Niklas Luhmann), for understanding OH as a system of functional sub-systems (human and animal health, governance levels), and second, for analyzing the degree of involvement in the collaborative work. This use of two notions of system is based on the two mentioned assumptions. |
P4. L5-14 |
|
|
L 104 – ‘samples’ – I presume you mean here the sampling for the interviewees ? The process of sampling is not described – what was the basis for the ‘purposive’ sampling; how large is the study population; how many were sampled; and how were those sampled identified ? |
Thanks, we have expanded the sampling strategies now. As there were very few actors present at the administrative level, we have approached all the actors of the human and the animal health system and most of them provided consent to participate in the study. Thus, for for the qualitative data collection, we have recruited almost all the administrative actors working at the AMC and purposively selected the lead non-governmental organizations and private bodies. Similarly, we have purposively also selected some actors from the provider level, till the saturation of responses. For the quantitative data collection, in addition to the above participants, we have sent an open invitation to all the clinicians working in the human and animal health system of the city, and those who agreed to participate (40% response rate) and who provided consent, were recruited for the survey. |
P4. L23-31 |
|
|
L 120 – how was the structured network questionnaire administered ? by interview or was it transmitted in some way ? |
The structured network questionnaire was administered personally by a trained research assistant, that we have mentioned. |
P4. L43-44 |
|
|
There is no mention of ethical review or ethical procedures in the data collection – although tis is noted in the notes at the end of the paper. |
Thanks. Now we have moved the ethics to the end of method section and mentioned a sentence- A written consent was collected from each participant, who were recruited in this study.
|
P6. L1-4 |
|
|
Results
|
Please provide a summary of the response rate and the key features of the informants – what positions; what sector |
Thanks for the suggestion. We have now removed the figure 1 and added table-1 with all these new information. We have also clubbed the information from the figure-2 in the newly developed table 1. |
P6. Table 1 |
|
L 150 – what is meant by ‘two-layered’ actors ? do you mean that actors were based in one of two layers – the AMC and the district administrative body ? what is the relationship between these bodies ? A diagram of the institutions involved and their relationships would assist in understanding. This would assist in understanding this paragraph. Figure 1 does not clarify the organizational relationships. |
The two layers of actors on the two administrative levels of district and city result in a strong influence from the district or even higher authorities on the action of zoonoses prevention in the city. As figure 1 was not intuitive enough, we decided to replace it with table 1. |
P6. Table 1 |
|
|
See comment above on providing more on the governance / organizational context. |
We have added the organizational context in the method section |
P3. L17-36 |
|
|
In what ways does the ‘outbreak’ situation differ from the non-outbreak situation – particularly in terms of governance and distribution of authorities ? Does an outbreak declaration change the distribution of authority ? This might help understand the differences found here. |
We have considered adding a sentence for better clarity- Although the governance structure remains the same in outbreak and in non-outbreak situations, the increased influence of the district/higher administrative actors are noteworthy during the outbreak situation. |
P6. L28-31 |
|
|
3.2 Interest-influence matrix The matrix could be described in the materials and methods section, as well as what is meant by the differentiation between players, subjects, context setters or crowd. This would assist in understanding this section. |
Thanks. We have described in details about the parameters of the matrix in the method section |
P5. L17-30 |
|
|
Line 185-186 – This comment and other comments what is ‘needed’ eg ‘ need to be strengthened with certain powers’ are more of an opinion / interpretation rather than a finding, and belong in the discussion section |
Yes, thus we have deleted these two lines |
P7. L17-19 |
|
|
Line 191-192 – Again a comment this is more an interpretation / recommendation – and should be shifted to the discussion. |
Yes, thus we have deleted these two lines |
P7. L20-21 |
|
|
Line 232 – ‘ power issues’ – please explain what is meant by this term. The next sentence talks about ‘who would lead’ – is this a question of ‘who has the authority to lead ?’ or is it a question of who is interested / motivated to lead ? |
Thanks. We have clarified now and edited the sentence as per your suggestion. There were different challenges for collaboration highlighted by the actors, one of the major challenge was who is interested or motivated to lead such action. |
P8. L35-37 |
|
|
Line 234 – what sort of ‘collaborative activities’ – what sort of collaborative activities are meant here ? |
We have replaced the term collaborative activities with collaborations |
P8. L38 |
|
|
Given the differences between outbreak and non-outbreak situation mentioned above – do these power issues relate to both situations ? or more in one than another ? |
The power issues are more at stake in non-outbreak situations, as during the outbreak the local government has to follow the strategies by the higher authorities (top-down directives). Thus, we have added this information now. As observed in the local context, collaborations only happen with top-down directives during outbreaks, the power issue, however, emerges during non-outbreak situations and leads to the questions what needs to be done who should do it |
P8. L29-31 |
|
|
Line 247 ‘were observed’ – by whom ? by the researchers or by the informants ? The comment regarding ‘need to streamline’ – is that a finding or an opinion of the researchers ? If so should be moved to the discussion. |
Were observed is replaced by were reported by the participants Yes, ‘need to streamline’ was a suggestion by the researcher thus deleted in the result section |
P9. L11-12 |
|
|
3.4 Network analysis The parameters discussed here and the analytic approach should be presented in the methods / materials and explained eg the degree of centralization and the average degree; the network figures – at least in broad terms. |
Thanks for the suggestion. We have now described the parameters of the network analysis in the method section. |
P5. L36-45 |
|
|
Line 289 Are there confidence intervals defined for these parameters ? how significant are the differences between the parameters between outbreak and non – outbreak situations? |
UCINET software could not generate the CI for these parameters thus, we have removed the term significantly |
P10. L3 |
|
|
Line 290 ‘reduced significantly’ – is this statistical significance or some other measure ? |
No, the term significant should not be used here. Thus, we have deleted the term significantly. |
P10. L13 |
|
|
Line 295 / Figure 3: please provide some explanation for the figures – what do the line s and the position of the different actors indicate ? |
Thanks for the suggestion. We have included below information as part of the figure explanation. Figure 1 & 2 represent the network cohesion during the outbreak and non-outbreak situation. The lines between the actors represent their interaction. , Whereas an arrow pointing away from an actor towards another actor shows that only the former imentioned the latter, a line with arrows at both ends is a mutual relationship. |
P13. L 1-4 |
|
|
Line 315: This is the first mention of the number of informants – as mentioned above, the response rate should be included at the start of the results section. |
Thanks, We have address the participant details at the beginning of the result section. |
P6. L7-9 |
|
|
Line 334 : was the ‘urging to shift ISC to the bottom level’ an opinion expressed by the informants, or is this the opinion of the researchers ? |
Yes, this is an information provided by the participant. We have clarified and edited the sentence now. …. thus these provider actors urged to shift the ISC to the bottom level. |
P13. L9-10 |
|
|
Discussion
|
Line 340: refers to prevention and control of zoonoses, rather than OH. Need to be consistent on whether this is about OH or zoonosis control. |
It is about OH actors. Thanks for pointing this out. We have replaced the term |
P14. L13 |
|
Line 342: ‘integral role in decision making’ – there is no mention of decision making in the results – need to justify this statement. |
We made it consistent now. As the term decision making was not in the result, we have replaced the term with influence. |
P14. L14 |
|
|
Line 346: what is the ‘firmer borderline’ and how does this relate to ISC ? |
Firmer borderline in the sense of the governance structure, as the higher authorities from the district/state have influenced the city actors towards collaborations during the outbreak. |
P14. L18-19 |
|
|
Line 349 – ‘powers devolved to them’ – there is no reporting of the powers devolved to local government bodies in the results section |
During the outbreak, the order for the collaborative actions comes from the top authorities, however the power remained with the higher authorities only. The similar findings we have reflected in the result section L 323-351 |
No change |
|
|
Line 352 – ‘dependent for required resources’ – again there is no report of this in the results section |
There are quotes justifying this statement L 365-366 & L 388-390 |
No change |
|
|
Line 358-359 – the issue of vertical collaboration (across levels) is important, but is only raised at this point. It could be mentioned in the introduction and methods as one of the elements for the analysis. |
Thanks for the suggestion. We were not aware of this phenomenon prior to the study. The data analysis indicated this finding, which are speculating as an important element for OH operationalization. |
No change |
|
|
Line 366 – while HSS may be ‘everybody’s business’, this research was not about HSS but about OH collaboration – so the relevance of the private sector to OH should be the basis for their involvement. |
Thanks for pointing this out. We have provided a literature reference and edited the sentence now. One Health Network analysis by Spencer J et al. indicated the minimal engagement of private for-profit sectors in OH [53]. In global health governance, the private and non-governmental actors contribute significantly towards improving health for all [54]. |
P14. L37-39 |
|
|
Line 380 – the distinction between outbreak and non-outbreak contexts is a key finding – more analysis of what changes in the context and in the relationship[s between outbreak and non outbreak situations would be a useful addition |
Influence from the context in the form of directives and additional resources is increased in outbreak situations and therefor also the relationship between actors more intense as shown by the network analysis. The context, however, was not objective of the study and can not be adequately analysed from the data. |
No change |
|
|
Line 384 ‘their leadership capacity’ – whose leadership capacity ? |
The leadership capacity of the city administrators |
P15. L9 |
|
|
Line 389 ff: useful discussion on provider level – are there differences at this level between outbreak and non outbreak situations ? |
We have added these information to the discussion- The network cohesion among the provider level actors didn’t differ much between the outbreak to the non-outbreak situation. One of the potential reasons could be the predominant presence of private veterinarians in the network, and the other reason could be that only few of the provider actors were engaged in collaborative activities during the outbreak which resulted into the minimal difference in the network cohesion at the provider level. |
P15. L30-35 |
|
|
Line 407 ff : these are a series of recommendations in how to improve ISC – and could include here some of the references to what is needed provided elsewhere in the paper. It would strengthen the recommendations to provide references as to where the proposed actions have been found to be effective esp lines 416 ff. |
Thanks for your valuable suggestion, now we have edited the same. |
P15. L46-50 P16. L5-7 |
|
|
Line 422 Limitations. It might be useful to note that the study was undertaken in one city, and that the generalizability of the findings needs to consider that specific context. Given these limitations, here generalizable are the findings ? |
We have added this now to limitation section- This is important to note that the study was undertaken in one city of India and that the generalizability of the findings might be constrained to any other part of the country as cities of India are with different governance structure. However, the approach used in this study could be utilized in any context for exploring OH actors. |
P16. L12-15 |
|
|
There is no discussion of how the findings of this study compare with other studies of OH collaboration either in India or elsewhere. |
Thanks for raising this issue! As per our knowledge there is no such study on OH collaboration in India or elsewhere focusing on different levels of the health system. However, we attempted to provide references or cite appropriate studies conducted OH collaboration only at the provider level or administrative level. |
No change |
|
|
Line 444 the recommendation for a OH taskforce is interesting – but only arises in the conclusion section. The study did not explore this option – at least it was not reported – so the basis for this recommendation is not clear. |
Thanks again for another valuable point. Yes, we do agree with your comment. However, we felt that this context is highly relying on the top down directive approach for the implementation. As per our previous scoping review (https://www.ncbi.nlm.nih.gov/pmc/articles/PMC6606562/), we suggested this recommendation as part of the third-party collaboration might work for this setting. We have made clear in the conclusions that a OHTF would require further studies. |
P16. L34, L42-44 |

Round 2
Reviewer 2 Report
The issues and concerns raised in the review of the first version of the paper have been satisfactorily addressed or an explanation provided; the additional information and references has strengthened the paper.
A couple of minor issues which the authors can address:
Need a check of syntax, and punctuation (eg line 39 p 15 uses Oh instead of OH)
The presentation of data as Table 1 is a useful improvement but suggest use the actual terms (crowd, player etc) rather than initials as the reader has to keep referring down to the legend at bottom of the table to check the abbreviated term
It is not clear what is meant by ‘edges’ line 37 page 5
P 14 line 13 ‘for the OH’ – syntax not clear. On reflection, the focus of the study is more on zoonosis prevention and control as a signal function for OH ; while ‘for the OH’ is not very clear. Suggest revert to the previous wording of ‘zoonosis prevention and control’.
Author Response
Dear Reviewer,
Thanks for your valuable time in reviewing the revised manuscript. We are immensely thankful for your comments and suggestions to improve this draft. We have addressed your minor comments as mentioned in below table-
|
Reviewer comments |
Author’s response |
Page & Line No. |
|
Need a check of syntax, and punctuation (eg line 39 p 15 uses Oh instead of OH) |
We have corrected and now it’s denoted as OH |
P15 L39 |
|
The presentation of data as Table 1 is a useful improvement but suggest use the actual terms (crowd, player etc) rather than initials as the reader has to keep referring down to the legend at bottom of the table to check the abbreviated term |
Thanks for the suggestion, We have edited the same in the table 1 |
P6 Table 1 |
|
It is not clear what is meant by ‘edges’ line 37 page 5 |
Thanks, we have added edges or links |
P5 L37 |
|
P 14 line 13 ‘for the OH’ – syntax not clear. On reflection, the focus of the study is more on zoonosis prevention and control as a signal function for OH ; while ‘for the OH’ is not very clear. Suggest revert to the previous wording of ‘zoonosis prevention and control’. |
Yes, we do agree. Thus the term OH is now replaced with previously mentioned term for the prevention and control of zoonoses. |
P14 L13 |
